# Multivariable Decoupling and Nonlinear Correction Method for Image-Based Closed-Loop Tracking of the Risley Prisms System

**DOI:** 10.3390/mi13122096

**Published:** 2022-11-28

**Authors:** Huayang Xia, Jinying Li, Yunxia Xia, Liangzhu Yuan, Wenxue Zhang, Haotong Ma, Piao Wen

**Affiliations:** 1Key Laboratory of Optical Engineering, Chinese Academy of Sciences, Chengdu 610209, China; 2Institute of Optics and Electronics, Chinese Academy of Sciences, Chengdu 610209, China; 3University of Chinese Academy of Sciences, Beijing 100049, China

**Keywords:** Risley prisms, nonlinear correction, multivariable decoupling, variable gain, image-based closed-loop

## Abstract

Image-based closed-loop tracking (IBCLT) is an important part of the process of target tracking. The Risley prism system has a unique advantage in improving the target tracking ability because of its compact and lightweight structure. Compared with traditional target tracking equipment, the Risley prism system has two difficulties in the process of IBCLT. First, the Risley prism is a complex coupling system of double input and double output. Second, the Risley prism itself is a nonlinear system. These problems lead to decrease in dynamic response and inconsistent target tracking capabilities. Thus, this paper proposes a method to implement multivariable decoupling and reduce the nonlinear effect. First, the boresight error of IBCLT is decoupled to the azimuth and elevation directions by the rotation matrix error-decoupling (RMED) method. Second, the gains of IBCLT in azimuth and elevation directions are independent variables that comes from two functions of the target elevation angle. The experimental results show that the IBCLT error deviation of different static targets in the field of view is within 0.025 arcsec, which is 70% lower compared with the fixed gain method. Furthermore, the steady-state error deviation of moving targets is controlled within 2.5 arcsec. These experimental results prove the feasibility and effectiveness of the proposed method.

## 1. Introduction

A Risley prism comprises two coaxially arranged prisms. An incident light beam can be refracted through the Risley prism to obtain different exit angles by changing the rotation angle of the prisms [1,2,3,4,5]. The Risley prism system is shown in Figure 1. Compared to traditional beam pointing devices, e.g., gimbal and fast mirror devices [6,7,8,9], the Risley prism benefits from a compact structure, high pointing accuracy, small size, good dynamic performance, and large aperture beam deflection. In addition, Risley prisms can be used for both beam scanning and target tracking. Currently, Risley prisms are used in various applications, e.g., military reconnaissance, infrared countermeasures, biomedicine, and laser radar [10,11,12,13,14].

In traditional target tracking devices, e.g., gimbal and fast mirror devices, the relationship between beam pointing and the mechanism deflection angle is typically linear; however, the relationship between the rotation angle of a Risley prism and beam pointing is nonlinear. When the target moves, the nonlinear relationship between the rotation angle and beam pointing is inconsistent. Thus, to realize fast and smooth target tracking, an effective target tracking control method that is based on an analysis of the nonlinear problem of Risley prisms must be employed.

Previous studies have investigated the nonlinear problem associated with Risley prisms. For example, in 2006, to eliminate the nonlinear problem in a beam control actuator for a Risley prism beam pointer device developed by Ball Aerospace and Technologies Corp, Ostaszewski proposed a local linearization method. However, the nonlinear problem and linearization method were not analyzed extensively [15]. In 2008, Yang proposed a reverse analysis method for the Risley prism system and provided nonlinear equations between beam pointing and the prism rotation angle [16]. In addition, Zhou obtained the nonlinear relationship between the rate of the prism rotation angle and the beam deflection rate through a theoretical analysis [17]. Anhu Li from Tongji University analyzed the nonlinear relationship between the emergent beam and the rotation angle of the prism based on a two-step method and the control singularity of the center and edge of the field of view of the Risley prism system [18]. Many previous studies have focused on the theoretical analyses of the nonlinear problem of the Risley prism. However, these studies did not investigate specific methods to reduce the nonlinear influence and achieve smooth target tracking. 

Our research group has studied the IBCLT theory of the Risley prism system and achieved high-precision tracking of the target. However, these studies have not fully eliminated the nonlinearity problem of the Risley prism system [19,20].

Thus, this paper proposes a nonlinear correction method to reduce the nonlinear influence and improve the dynamic response capability of the Risley prism system when tracking a target. Here, designing the gain of the IBCLT as a function of the target elevation angle achieves smooth target tracking. In addition, the proposed method can be used to track both static and moving targets.

The remainder of this paper is organized as follows. In Section 2, the nonlinear relationship of the Risley prism is analyzed. In Section 3, the proposed nonlinear control method for the Risley prism system is introduced. Section 4 describes a simulation of the proposed method, and Section 5 describes an experimental evaluation. Finally, conclusions are presented in Section 6.

## 2. Nonlinear Analysis of the Risley Prism System

The field of view of the Risley prism system is divided into four regions. In Figure 2, region I represents the blind area of the field of view of the Risley prism system. Region II and region IV represent the large nonlinear region. In region IV, the increase of elevation angle leads to the enhancement of nonlinearity. In region II, as the elevation angle decreases, the azimuth angle with the same variation requires a faster rotation speed. This is the reason why the nonlinearity in this region becomes larger. Region III is the small nonlinear influence of the system. $\Phi_{m}$ is the Risley prism system dynamic response to the best viewing area to track the target. 0°–360° is the azimuth angle.

This section analyzes the nonlinear relationship between the Risley prism’s rotation rate and the target’s position.

According to an analysis performed using the first-order paraxial approximation method [21], the deflection angles $\delta_{1}$ and $\delta_{2}$ of a beam refracted through prism1 and prism2, respectively, are only related to the prism apex angles $\alpha_{1}$ and $\alpha_{2}$ and refractive indexes $n_{1}$ and $n_{2}$:(1)$$\delta_{1}=\alpha_{1}(n_{1}-1)$$
(2)$$\delta_{2}=\alpha_{2}(n_{2}-1)$$

To facilitate the analysis of nonlinearity, it is assumed in Section 2 that the apex angles $\alpha_{1}$, $\alpha_{2}$ and refractive indexes $n_{1}$, $n_{2}$ of prisms1 and prisms 2 are the same.

To simplify the analysis of the relationship between the prism’s rotation rate and the target position, we divide the target into radial motion and tangential motion.

### 2.1. Radial Motion

According to the first-order paraxial approximation method [21], the relationship between the target elevation angle Φ and relative rotation angle Δθ can be obtained as follows.
(3)$$\Phi =\sqrt{\delta_{1}{}^{2}+\delta_{2}{}^{2}+2\delta_{1}\delta_{2}\cos\left(\Delta \theta \right)}$$

According to (3), when the target moves radially, i.e., the elevation angle of the target changes and the azimuth angle of the target remains unchanged, the relationship between the relative rotation angle rate of the two prisms and the elevation angle of the target can be obtained as follows.
(4)$$\frac{d\Delta \theta }{d\Phi }=\frac{1}{\delta \sqrt{1-{\left(\frac{\Phi }{\Phi_{\max}}\right)}^{2}}}$$
(5)$$\Phi_{\max}=2\delta$$

Here, $\Phi_{\max}$ is the maximum elevation angle of the field of view of the Risley prism system. In (4) and (5), $\delta =\delta_{1}=\delta_{2}$. The specific meaning of $\frac{d\Delta \theta }{d\Phi }$ is the slope of the relation curve between the relative rotation angle Δθ and the elevation angle Φ.

### 2.2. Tangential Motion

When the target moves tangentially, i.e., the elevation angle of the target does not change but the azimuth of the target changes. According to a previous study [22], when the target only moves tangentially, the ratio of the rotation angle rate of the Risley prism to the change rate of the target position is expressed as follows:(6)$$D_{1}=D_{2}=\frac{1}{\sin\Phi }$$
where *D*_1_ is the ratio of the rotation angle rate of prism1 to the change rate of the target position, and *D*_2_ is the ratio of the rotation angle rate of prism2 to the change rate of the target position. 

From (4) and (6), regardless of whether the target is moving radially or tangentially, the rotation angle rate of the prism is nonlinear with the elevation angle of the target.

## 3. Image-Based Closed-Loop Tracking of Risley Prism System

### 3.1. Image-Based Closed Loop Tracking

The IBCLT system of the Risley prism is shown in Figure 3. In the analysis of the IBCLT of the Risley prism system, the boresight error on the detector is decoupled to tracking errors in the azimuth and elevation directions using the RMED method. Decomposing the boresight error into the azimuth and elevation directions can simplify the multiple inputs and multiple outputs of the entire IBCLT system into two single-input and single-output systems.

CCD is an image detector. R is the expected imaging position of the target on the image detector. C represents the actual imaging position of the target on the image detector. C is obtained by the forward solution. The information of C is received through the image detector (CCD). The difference between R and C is the target boresight error E. The boresight error E includes the *X*-axis tracking error Img_x and the *Y*-axis tracking error Img_y in the Cartesian coordinate system. Tracking errors Img_x and Img_y are coupled with the rotation angles of prism1 and prism2, which need to be decoupled. $G_{s}\left(s\right)$ is the output characteristic of the image detector. ΔΘ is the tracking error in the azimuth direction, and ΔΦ is the tracking error in the elevation direction. ΔΘ and ΔΦ are obtained by decoupling the tracking errors Img_x and Img_y. $K_{A}$ and $K_{E}$ are the gains of the IBCLT in the azimuth directions and elevation direction. The key to the nonlinear correction method is the set up of $K_{A}$ and $K_{E}$. $G_{1}\left(s\right)$ and $G_{2}\left(s\right)$ are two proportional integral controllers in the azimuth direction and elevation direction. $K_{A}^{\prime }$ and $K_{E}^{\prime }$ represent the characteristics of azimuth direction and elevation direction of the controlled object (Risley prism system), which are the sources of the nonlinearity of the Risley prism system. $\theta_{1}$ and $\theta_{2}$ are rotating angles of prism1 and prism2.

#### 3.1.1. Multivariable Decoupling Method

The Multivariable decoupling method adopts the RMED method to decompose the detector boresight error into azimuth and elevation direction tracking errors [20]. The RMED method is shown in Figure 4.

Point O is the center point of the image detector and the target closed-loop point. $A\left(\Phi_{0},\Theta_{0}\right)$ is the imaging point of the target on the detector, which is the boresight error of the target. $B\left(\Phi_{t},\Theta_{t}\right)$ represents the target guiding position, and $C\left(\Phi_{m},\Theta_{m}\right)$ represents the target position synthesized by points A and B, which is the actual guiding position of the target. 

The REMD method converts the boresight error A under the coordinate system Oxy to the coordinate system $Ox^{\prime }y^{\prime }$. Then, the boresight error A under the Cartesian coordinate system $Ox^{\prime }y^{\prime }$ is converted to the polar coordinates, which can obtain the tracking error of the azimuth direction and elevation direction, respectively, and achieve boresight error decoupling.

#### 3.1.2. Nonlinear Correction of Elevation Loop

From (3), we find that the elevation angle of the beam after refraction through the Risley prism system is related to the relative rotation angle ∆θ. In addition, the relationship between the elevation angle of the beam and the relative rotation angle ∆θ is nonlinear, and the correlation coefficient $K_{E}^{\prime }$ is obtained as follows (assuming the apex angles $\alpha_{1}$, $\alpha_{2}$ and refractive indexes $n_{1}$, $n_{2}$ of prisms1 and prisms2 are the same).
(7)$$K_{E}^{\prime }=\frac{d\Phi }{d\Delta \theta }=|\frac{\sqrt{2}\delta_{1}\times \sin\Delta \theta }{2\sqrt{1+\cos\Delta \theta }}|$$

From (7), we find that the elevation loop is nonlinear. Here, when the elevation angle increases, the nonlinearity enhancement and the relative rotation angle rate of the prisms’ increase. The relative rotation angle rate of the prisms’ increase will lead to the tracking error increase. Thus, to maintain the consistent dynamic response capability of the elevation loop, the gain $K_{E}$ of the elevation loop must be designed as a variable gain to adapt to the nonlinearity of the elevation loop.

#### 3.1.3. Nonlinear Correction of Azimuth Loop

According to the first-order paraxial approximation method, the relationship between the azimuth angle of the beam and the rotation angle of the two prisms can be obtained as follows [21].
(8)$$\tan\Theta =\frac{\delta_{1}\sin\theta_{1}+\delta_{2}\sin\theta_{2}}{\delta_{1}\cos\theta_{1}+\delta_{2}\cos\theta_{2}}$$

Note that the apex angle and refractive index requirements of the two prisms in the Risley prism system are the same; thus, (8) can be simplified as
(9)$$\tan\Theta =\tan[(\frac{\theta_{1}+\theta_{2}}{2})]$$

The azimuth angle of the beam is related to the sum of the rotation angles of the two prisms. This is a linear relationship, and the correlation coefficient $K_{A}^{\prime }$ is obtained as follows.
(10)$$K_{A}^{\prime }=\frac{1}{2}$$

In contrast to the elevation loop, the azimuth loop is linear; however, when the target only moves in the tangential direction, the azimuth angle of the field of view changes inconsistently when the target moves the same distance at a different elevation angle in the field of view. From Figure 5, when target 1 moves from point $A_{1}$ to point $B_{1}$ in the field of view area III, the azimuth of the field of view changes is $\Delta \Theta_{1}$, and when target 2 moves from point $A_{2}$ to point $B_{2}$ in the field of view area II, the azimuth of the field of view changes is $\Delta \Theta_{2}$. The elevation angle of point $A_{1}$ is equal to the elevation angle of point $B_{1}$. The elevation angle of point $A_{2}$ is equal to that of point $B_{2}$. The elevation angle in the field of view area III is greater than the elevation angle in the field of view area II. And the distance from point $A_{1}$ to point $B_{1}$ is equal to the distance from point $A_{2}$ to point $B_{2}$. However, the change of azimuth $\Delta \Theta_{1}$ is not equal to the change of azimuth $\Delta \Theta_{2}$, and $\Delta \Theta_{2}$ is greater than $\Delta \Theta_{1}$. The above analysis shows the greater zimuth angle of the field of view changes with the target moving the same distance when the target makes a tangential motion at the position with the smaller elevation angle of the field of view. In addition, the greater the angle of synchronous rotation of the two prisms, the greater the tracking error ΔΘ of the target is. Thus, the gain $K_{A}$ of the azimuth loop should also be designed as a variable gain to adapt to the error change of the azimuth loop.

### 3.2. Nonlinear Control of Risley Prism System

Combined with the nonlinear analysis and IBCLT analysis results of the Risley prism system, we conclude that the relative rotation rate of the prism increases when the target is in region IV, and the ratio of the prism rotation rate to the target position change rate increases when the target is in region II. In regions IV and II, the nonlinearity of the system increases, and the dynamic response to the tracked target decreases. Thus, to achieve smooth tracking, the system must respond rapidly when the target is located in regions IV and II.

Based on the above analysis, this paper proposes a method to reduce the effect of nonlinearity on target tracking in the Risley prism system. Specifically, the gains $K_{A}$ and $K_{E}$ in the IBCLT are designed as a function with the target elevation angle as an independent variable. $K_{A}$ and $K_{E}$ are adjusted automatically with the change in the target elevation angle. As a result, the gain of the IBCLT can be increased when the target is located in regions II and IV, where the nonlinearity enhancement and prism rotation rate increase. This effectively speeds up the response and reduces the tracking error. When the target is located in regions III where nonlinearity is weak, the gain of the IBCLT is controlled to ensure the dynamic performance of the system. In the proposed method, $K_{A}$ and $K_{E}$ are designed as the following functions.
(11)$$K_{A}=K_{1}+K_{2}\frac{1}{\sin\Phi }$$
(12)$$K_{E}=K_{3}\frac{1}{\sqrt{1-{\left(\frac{\Phi }{\Phi_{\max}}\right)}^{2}}}$$

Here, (11) and (12) are derived from (4) and (6), and $K_{1}$, $K_{2}$, and $K_{3}$ are constants. Different Risley prism systems have different values of $K_{1}$, $K_{2}$, and $K_{3}$. In the experiment, ten targets at different positions were tracked in the IBCLK. $K_{A}$ and $K_{E}$ values of the above ten targets were recorded when the tracking error is within 0.05 arcsec. The values of $K_{1}$, $K_{2}$, and $K_{3}$ are obtained by fitting these ten $K_{A}$ and $K_{E}$ values.

## 4. Simulation Analysis

The simulation model of the variable gain IBCLT system is illustrated in Figure 6.

The simulation system is made up of eight modules. Module 1 is the simulation time. Module 2 is the target position. The Φ is the elevation angle and Θ is azimuth angle of target. Module 3 is the inverse solution of the Risley prism system. The $\theta^{*}$ represents the rotation angle of the Risley prism system after inverse solution. Module 4 is an adder function unit. Its function is to add the prism rotation angles obtained from the boresight error to the prism rotation angles calculated from the target position. Module 5 is a servo control system. The module is a simulation of the motor characteristics in the system. Module 6 is the forward solution in the Risley prism system. The u in Module 6 is a complex coefficient which contains the actual rotating angles of prism1 and prism2 controlled by the servo system and target position. The E in Module 6 is a boresight error that includes the *X*-axis tracking error Img_x and the *Y*-axis tracking error Img_y in the Cartesian coordinate system. The actual imaging position of the target on the image detector is obtained by the forward solution. Boresight error E can be obtained by making the difference between the actual imaging position and the expected imaging position. Module 7 is the IBCLT unit. The module includes the core content of this paper, which are multivariable decoupling and nonlinear correction method. The input of Module 7 includes tracking errors Img_x and Img_y, target position, and simulation time. The output of Module 7 are the tracking errors in azimuth direction and elevation direction. Module 8 are two proportional-integral (PI) controllers that control the tracking errors in the azimuth and elevation directions. 

The different positions of the target in the field of view during the simulation are shown in Figure 7.

Ten different target locations, i.e., T1–T10, were selected from regions II, III, and IV (T1–T3 are located in region II, T4–T9 are located in region III, and T10 is located in region IV). Note that region I is the blind area of the system’s field of view; thus, it is not considered in this paper.

In the simulation, fixed gain control and variable gain control methods were used for the gain of the IBCLT to track different targets. The fixed gain control is expressed as follows.
(13)$$K_{A}=K_{E}=K^{\prime }$$

In addition, the variable gain control settings are given as follows.
(14)$$\{\begin{array}{cc} K_{A}=K_{1}+K_{2}\frac{1}{\sin\Phi } & \Phi \geq \Phi_{1}\\ K_{A}=K_{A}{}_{\left(\Phi_{1}\right)} & \Phi <\Phi_{1} \end{array}$$
(15)$$\{\begin{array}{cc} K_{E}=K_{E}{}_{\left(\Phi_{2}\right)} & \Phi >\Phi_{2}\\ K_{E}=K_{3}\frac{1}{\sqrt{1-{\left(\frac{\Phi }{\Phi_{\max}}\right)}^{2}}} & \Phi \leq \Phi_{2} \end{array}$$

$\Phi_{1}$ and $\Phi_{2}$ are the minimum and maximum target elevation angles, which can ensure system stability by changing the gain in the IBCLT.

The simulation results are shown in Figure 8.

The tracking error is the boresight error obtained by the image detector. The tracking error can be calculated by the square root of the x-direction tracking error Img_x and y-direction tracking error Img_y in the image detector as follows.
(16)$$\mathrm{error}=\sqrt{\mathrm{Img}\_x^{2}+\mathrm{Img}\_y^{2}}$$

As shown in Figure 8a, when the gain is fixed in the IBCLT, the IBCLT ability of the target at different positions in the field of view is inconsistent. In addition, when the target is in regions II and IV, the tracking error increases, and the tracking ability decreases. As shown in Figure 8b, the IBCLT ability of the target with different positions is effectively the same when the variable gain is employed in the IBCLT.

As shown in Table 1, when the gain is fixed in the IBCLT, the average tracking errors in different areas of the field of view differ significantly. Note that the average tracking error of target T1 in region II is 16 times that of target T6 in region III, and the average tracking error of target T10 in region IV is five times that of target T6 in region III. When the gain is variable in the IBCLT, the average tracking errors of the different targets exhibit little difference, and the maximum tracking error is less than three times. For example, target T10 has a maximum average tracking error of 0.076 arcsec, and target T8 has a minimum tracking error of 0.026 arcsec. According to the simulation results, we found that the proposed method realizes smooth target tracking in different fields of view.

## 5. Target Tracking Experiment 

Figure 9 shows the experimental platform, including the lamp, collimator, fast steering mirror, achromatic Risley prism system, and image detector. In this experiment, the target position is determined by the lamp, the collimator, and the fast steering mirror. Note that the position of the lamp is fixed in this experimental environment. After the light beam is collimated through the collimator, different positions of the outgoing light beam can be obtained by adjusting the deflection angle of the fast steering mirror such that different target positions can be simulated. The light beam is refracted through the achromatic Risley prism system and then imaged on the image detector. The boresight error in the image detector is then extracted and decoupled using the RMED method. In addition, the proposed method is employed to achieve smooth target tracking in the IBCLT.

The achromatic Risley prism system in the experimental platform is based on the model in literature [23]. The achromatic Risley prism system is shown in Figure 10, where prism 1 is composed of prism 101 and prism 102 and prism 2 consists of prism 201 and prism 202.

The solution relation of the achromatic Risley prism is similar to that of the Risley prism shown in Figure 1. The difference is that the refraction angle $\delta_{1}$ and $\delta_{2}$ of the achromatic Risley prism is as follows:(17)$$\delta_{1}=\alpha_{1}\left(n_{1}-1\right)-\alpha_{11}\left(n_{11}-1\right)$$
(18)$$\delta_{1}=\alpha_{2}\left(n_{2}-1\right)-\alpha_{22}\left(n_{22}-1\right)$$

The deflection angle of the fast mirror is $\pm 1.5^{{}^{\circ}}$, the field of view of the achromatic Risley prism is $\pm 3^{{}^{\circ}}$, and the field of view of the image detector is $0.2^{{}^{\circ}}$. The entire system was implemented on a PowerPC processor, including communication, servo control of the achromatic Risley prism system, IBCLT control, and data transmission (send and receive).

In this experimental platform, the servo control transfer function $G_{e}\left(s\right)$ of the Risley prism system is obtained as follows.
(19)$$G_{e}\left(s\right)=e^{-0.005s}\frac{2.4\times 10^{-8}s^{4}+1.1\times 10^{-5}s^{3}+1.8\times 10^{-3}s^{2}+0.11s+1}{8\times 10^{-18}s^{8}+1.1\times 10^{-14}s^{7}+7.2\times 10^{-12}s^{6}+3.4\times 10^{-9}s^{5}+8.9\times 10^{-7}s^{4}+1.7\times 10^{-4}s^{3}+5.9\times 10^{-3}s^{2}+0.11s+1}$$

The $G_{1}\left(s\right)$ and $G_{2}\left(s\right)$ is designed as the proportional-integral controller as follows.
(20)$$G_{1}\left(s\right)=G_{2}\left(s\right)=\frac{0.05s+0.992}{s}$$

In addition, the detector output characteristic $G_{s}\left(s\right)$ is expressed as follows.
(21)$$G_{s}\left(s\right)=e^{-0.06s}$$

The system transfer function $G_{E}\left(s\right)$ of the elevation loop is given as follows.
(22)$$G_{E}\left(s\right)=\frac{G_{s}G_{1}G_{e}K_{E}K_{E}^{\prime }}{1+G_{s}G_{1}G_{e}K_{E}K_{E}^{\prime }}$$

The system transfer function $G_{A}\left(s\right)$ of the azimuth loop is expressed as follows.
(23)$$G_{A}\left(s\right)=\frac{G_{s}G_{1}G_{e}K_{A}K_{A}^{\prime }}{1+G_{s}G_{1}G_{e}K_{A}K_{A}^{\prime }}$$

### 5.1. Static Target Tracking Experiment

As shown in Figure 11, nine different static target locations (a1–a9) in the achromatic Risley prism system were simulated by adjusting the fast steering mirror deflection angle.

The fixed gain is expressed as follows.
(24)$$K_{A}=K_{E}=K^{\prime }=0.2$$

Here, $K^{\prime }$ is the maximum fixed gain that can ensure system stability in the IBCLT within fields of view II–IV of the experimental platform.

According to (14) and (15), the parameters in the variable gain were set as follows.
(25)$$\{\begin{array}{c} K_{1}=0.11\\ K_{2}=0.075\\ K_{3}=0.186 \end{array}$$

Figure 12 shows the tracking errors of the different static targets using the fixed and variable gain techniques.

Figure 13 shows the frequency domain characteristics of the achromatic Risley prism system using the fixed gain control and variable gain control techniques when tracking target $a_{9}$.

Table 2 shows the dynamic response rising time $t_{rg}$ (i.e., the time required to reduce the initial tracking error from 90% to 10%) with fixed gain and rising time $t_{rv}$ with variable gain for nine different static target positions in the IBCLT control.

Figure 14 shows the dynamic response time distribution of the nine targets obtained using the fixed gain control and variable gain control techniques in the IBCLT.

Figure 12, Figure 13 and Figure 14 and Table 2 show that when the target is located at the inner and outer edges of the achromatic Risley prism field of view, nonlinearity is enhanced. In this case, the dynamic response capability of the fixed gain control technique is weakened. For example, the IBCLT bandwidth of target $a_{4}$ near $\Phi_{m}$ is 0.2709 Hz, whereas the IBCLT bandwidth of the target $a_{9}$ at the outer edge of the field of view is only 0.088 Hz. In addition, the dynamic response time of the IBCLT increases with fixed gain in the nonlinearity enhancement region.

The dynamic response capability of the region with strong nonlinearity can be increased using variable gain control. For example, Figure 13b shows that the IBCLT bandwidth of target $a_{9}$ at the outer edge of the field of view is increased to 0.2716 Hz when using variable gain control. In addition, the dynamic response time of the IBCLT of the target in the area with large nonlinearity is reduced. The dynamic response capability is improved by three times, and the deviation in tracking accuracy in the entire field of view is guaranteed to be within the 0.025 arcsec range. Thus, smooth target tracking is realized in the field of view of an achromatic Risley prism. 

### 5.2. Moving Target Experiment

In this experiment, three moving targets were simulated via the sinusoidal motion of the fast steering mirror, and the following moving targets were simulated.
(26)$$\{\begin{array}{cc} 0.13^{{}^{\circ}}+0.1^{{}^{\circ}}\sin(0.02\pi ) & \mathrm{target}1\\ 1.2^{{}^{\circ}}+0.1^{{}^{\circ}}\sin(0.02\pi ) & \mathrm{target}2\\ 2.7^{{}^{\circ}}+0.1^{{}^{\circ}}\sin(0.02\pi ) & \mathrm{target}3 \end{array}$$

In this experiment, target 1 moved in the achromatic Risley prism field of view in region II, target 2 moved in the achromatic Risley prism field of view in region III, and target 3 moved in the achromatic Risley prism field of view in region IV.

The parameter settings of the IBCLT for a moving target in the achromatic Risley prism system are consistent with those of a static target, and the formula for the IBCLT error for a moving target is also consistent with that of a static target. The fixed gain control and variable gain control techniques were used for the IBCLT of targets 1, 2, and 3. The results are shown in Figure 15.

In addition, the steady-state tracking error of IBCLT of the three moving targets is shown in Figure 16.

The steady-state error values of the IBCLT for the three moving targets are given in Table 3.

According to the results shown in Figure 15 and Figure 16, and Table 3, when the fixed gain control technique is employed for IBCLT of a moving target, the steady-state tracking error of the target in the achromatic Risley prism field of view in regions II and IV increases, and the steady-state tracking error deviation of the target tracking in different regions also increases.

The steady-state tracking error of regions II and IV can be reduced effectively using variable gain in IBCLT, and the steady-state tracking error can be controlled within 7 arcsec. In addition, the deviation of the steady-state error when tracking moving targets in different regions is within 2.5 arcsec.

## 6. Conclusions

In this paper, a method to implement multivariable decoupling and reduce the effect of nonlinear factors of the Risley prism system was proposed. Based on the IBCLT, the boresight error is decoupled into azimuth and elevation errors by using the RMED method. The IBCLT gain of the azimuth loop and the elevation loop comes from a function of target elevation angle. The experimental results demonstrate that the dynamic response capability of a static target in large nonlinearity region can increase three times. The tracking error deviation for different targets is within 0.025 arcsec when the variable gain control technique is used in the IBCLT. The IBCLT of moving targets ensures that the steady-state tracking error of different targets is controlled within 7 arcsec, and the steady-state tracking error deviation is controlled within 2.5 arcsec. The feasibility of the proposed method was verified theoretically and experimentally. Thus, we believe that the proposed method can reduce the effect of nonlinear problem of target tracking and achieve smooth target tracking within the field of view of the Risley prism system.

However, we did not consider the effect of the blind field of the Risley prism on the IBCLT of the target, which is another significant limitation. Thus, we plan to address this issue in future work.

## Figures and Tables

**Figure 1 micromachines-13-02096-f001:**
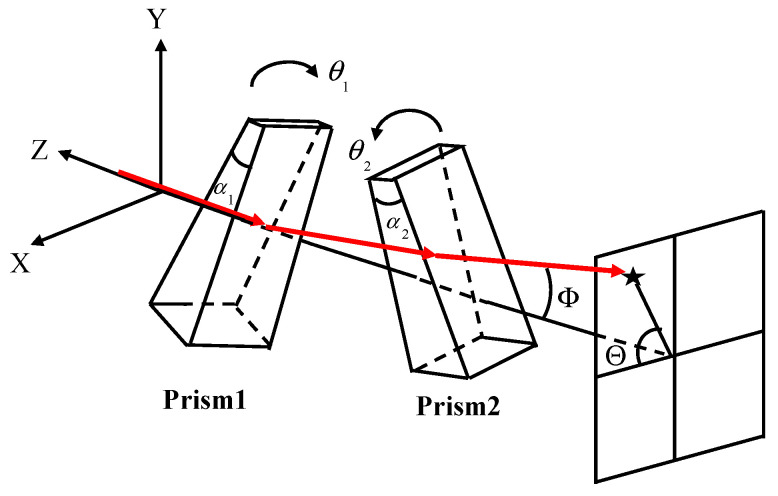
Risley prism system schematic. $\alpha_{1}$ and $\alpha_{2}$ are the apex angles of prism1 and prism2, $\theta_{1}$ and $\theta_{2}$ are the rotation angles of prism1 and prism2, Θ is the azimuth angle, and Φ is the elevation angle.

**Figure 2 micromachines-13-02096-f002:**
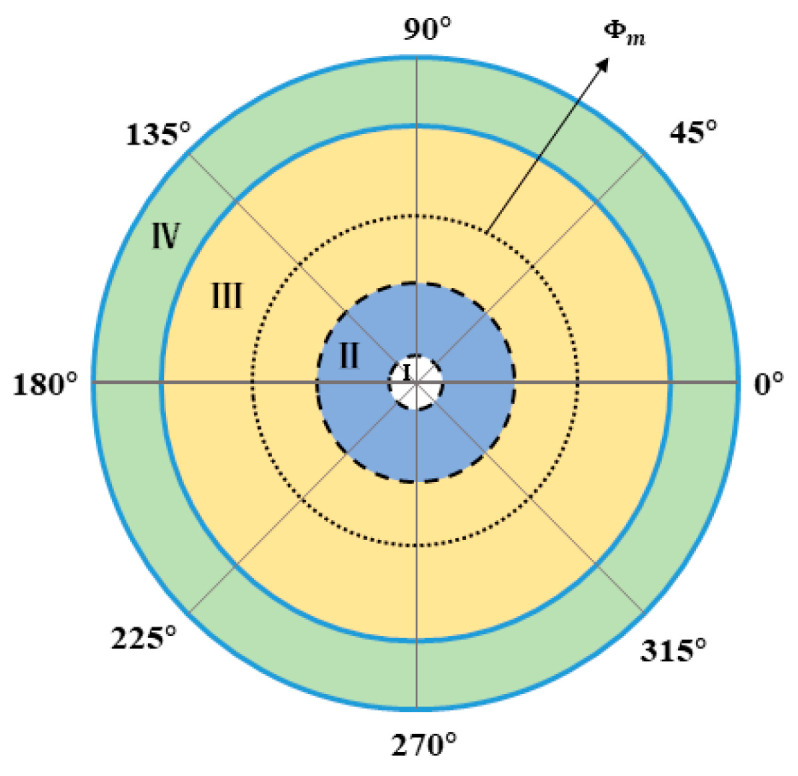
Field distribution of a Risley prism.

**Figure 3 micromachines-13-02096-f003:**
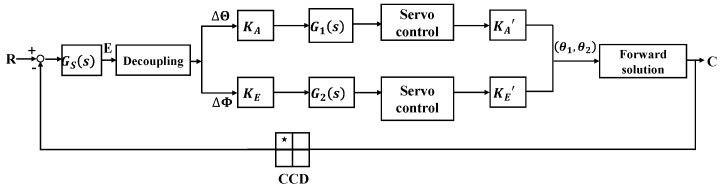
Schematic diagram of IBCLT for the Risley prism.

**Figure 4 micromachines-13-02096-f004:**
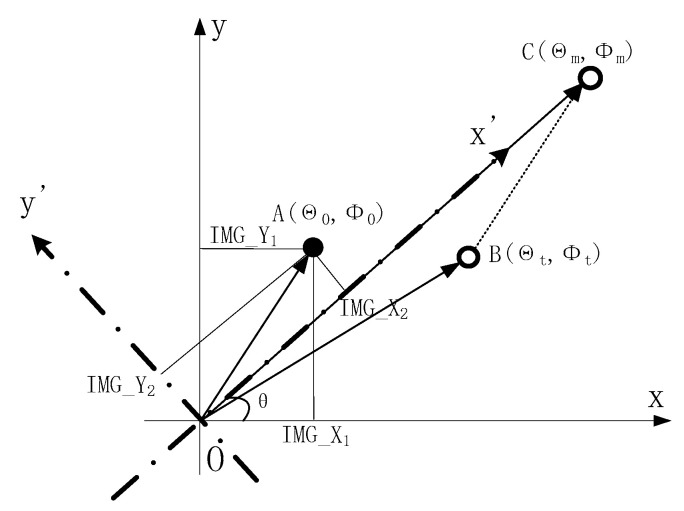
Projection diagram of boresight error and target position.

**Figure 5 micromachines-13-02096-f005:**
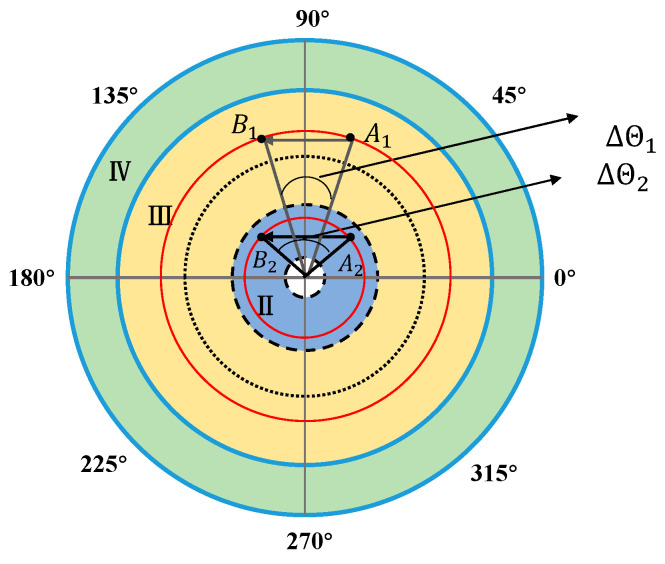
Comparison of azimuth angle changes.

**Figure 6 micromachines-13-02096-f006:**
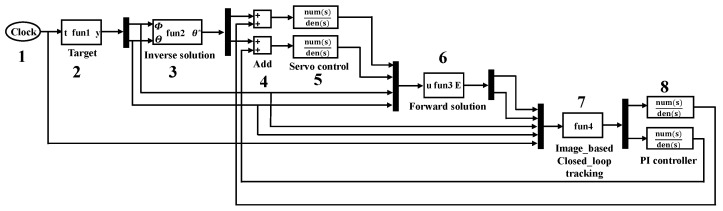
IBCLT control model diagram of the Risley prism system.

**Figure 7 micromachines-13-02096-f007:**
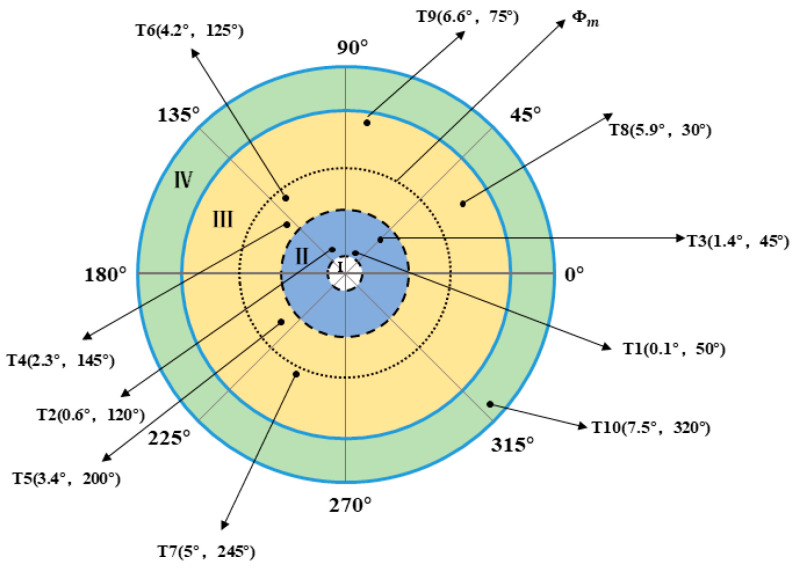
Different target positions.

**Figure 8 micromachines-13-02096-f008:**
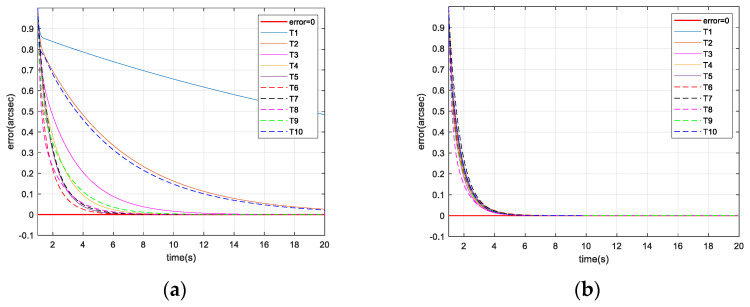
Tracking error of different target positions: (**a**) tracking error controlled by fixed gain and (**b**) tracking error controlled by variable gain.

**Figure 9 micromachines-13-02096-f009:**
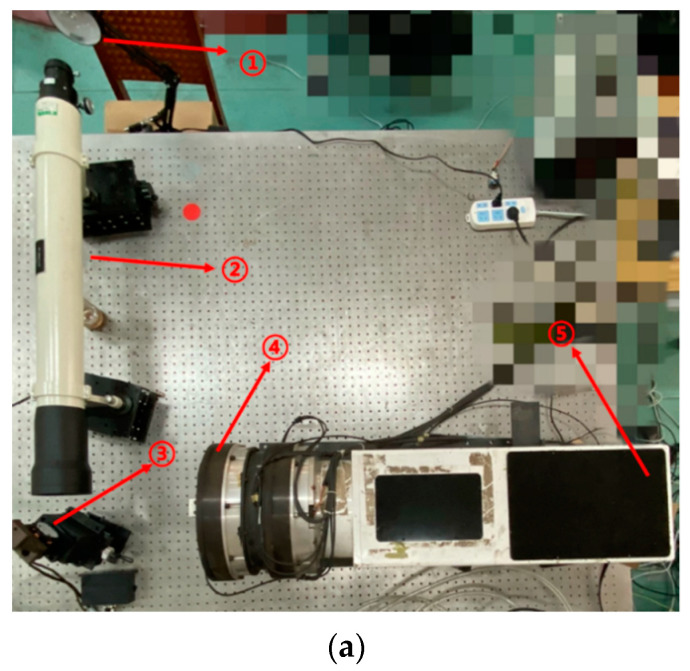
Experimental platform. (**a**) Main components of the experimental platform: ① lamp, ② collimator, ③ fast steering mirror, ④ achromatic Risley prism system, and ⑤ image detector. (**b**) Optical propagation scheme of the experimental platform.

**Figure 10 micromachines-13-02096-f010:**
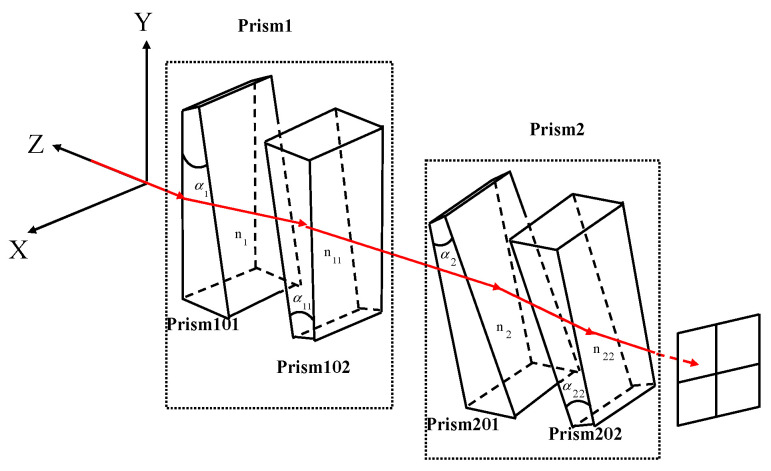
Schematic of an achromatic Risley prism system. $\alpha_{1}$, $\alpha_{11}$, $\alpha_{2}$, $\alpha_{22}$ are prism101, prism102, prism201, prism102 apex angles. $n_{1}$, $n_{11}$, $n_{2}$, $n_{22}$ are prism101, prism102, prism201, prism102 refractive indexes.

**Figure 11 micromachines-13-02096-f011:**
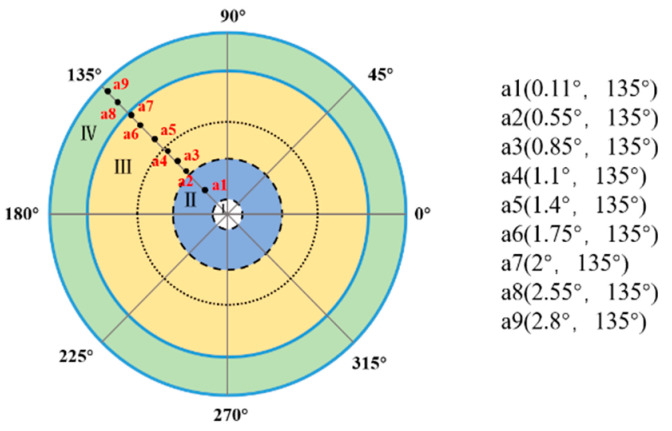
Nine simulated target positions.

**Figure 12 micromachines-13-02096-f012:**
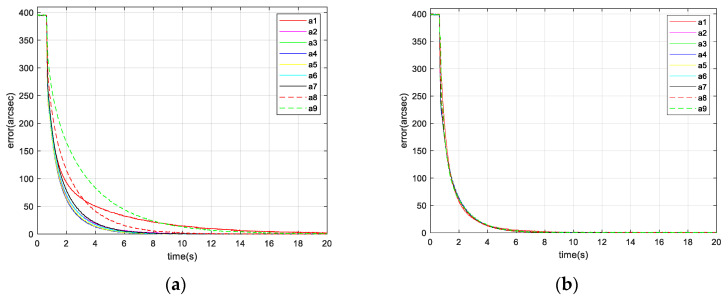
IBCLT errors of different static targets: (**a**) fixed gain and (**b**) variable gain.

**Figure 13 micromachines-13-02096-f013:**
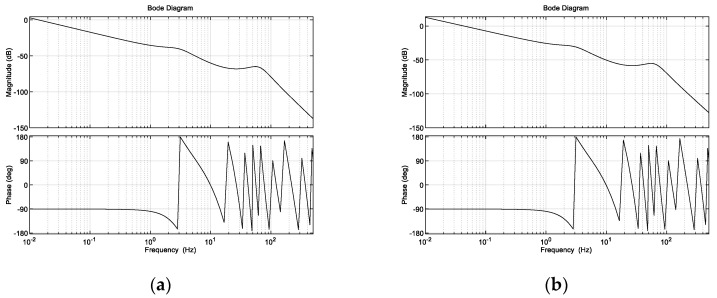
Frequency domain characteristic curve obtained with (**a**) fixed gain control and (**b**) variable gain control techniques.

**Figure 14 micromachines-13-02096-f014:**
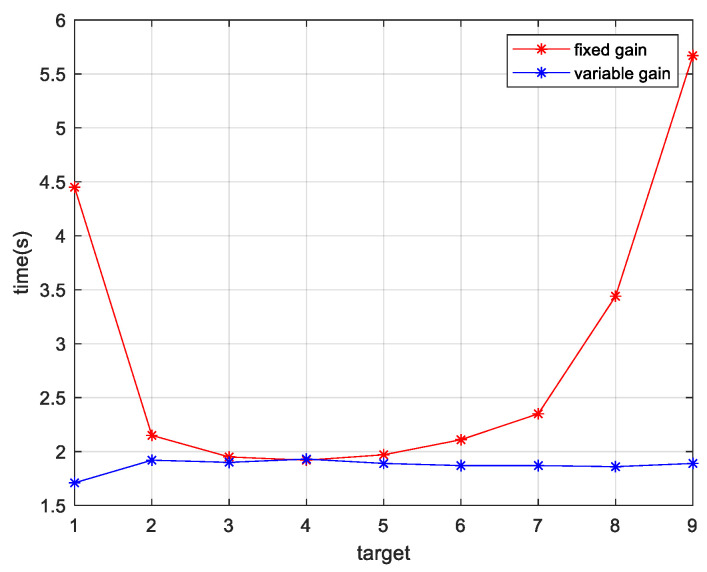
Dynamic response time distribution of IBCLT of nine different targets.

**Figure 15 micromachines-13-02096-f015:**
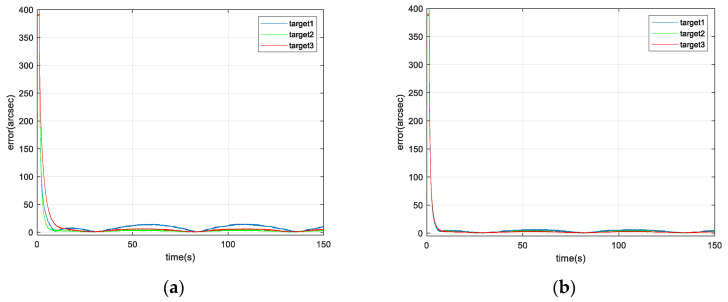
IBCLT error of different moving targets obtained with (**a**) fixed gain control and (**b**) variable gain control techniques.

**Figure 16 micromachines-13-02096-f016:**
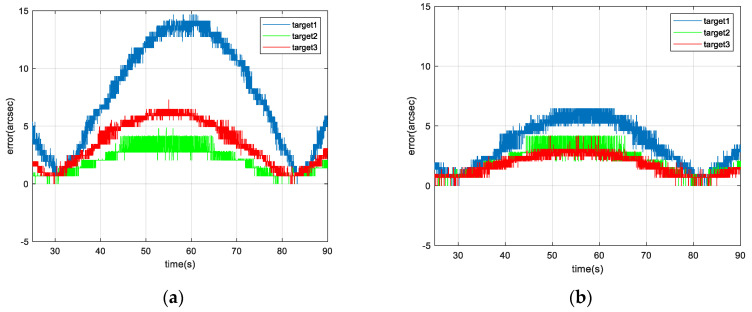
Local amplification of steady-state error in IBCLT of different moving targets: (**a**) fixed gain control and (**b**) variable gain control.

**Table 1 micromachines-13-02096-t001:** Average tracking errors of different targets.

Target	Fixed Gain (arcsec)	Variable Gain (arcsec)
T1	0.66	0.042
T2	0.24	0.034
T3	0.093	0.03
T4	0.059	0.032
T5	0.046	0.038
T6	0.039	0.039
T7	0.049	0.04
T8	0.039	0.026
T9	0.059	0.031
T10	0.22	0.076

**Table 2 micromachines-13-02096-t002:** Rising time of dynamic response for closed-loop tracking of different target.

Φ(°)	0.11	0.55	0.85	1.1	1.4	1.75	2	2.55	2.8
$t_{rg}$(s)	4.45	2.15	1.95	1.92	1.97	2.11	2.35	3.44	5.67
$t_{rv}$(s)	1.71	1.92	1.9	1.93	1.89	1.87	1.87	1.86	1.89

**Table 3 micromachines-13-02096-t003:** Steady-state tracking errors of different moving targets.

Gain	Fixed	Variable
target 1(arcsec)	14.6596	6.4623
target 2(arcsec)	4.8437	4.1667
target 3(arcsec)	7.2816	4.11

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
