# Peer review of "Multivariable Decoupling and Nonlinear Correction Method for Image-Based Closed-Loop Tracking of the Risley Prisms System"

_micromachines, 2022, doi:10.3390/mi13122096_

Round 1
Reviewer 1 Report
General comments:
1. Spelling, grammar, punctuation and text formatting need to be revised before manuscript is of publication quality. Some examples are:
2. Symbols in figures captions and in the text, seems to be displaced over the writing line.
3. Line 70, should be analyzed.
4. Line 71, should be describes.
5. Line 90, should be prism1 and prism2.
Particular comments:
6. Figure 3 shows the relationship between the rate of the relative rotation angle of the 103
7. Risley prism and the pitch angle of the target.
8. Figure 5 may be described in detail. What is the block in the feedback loop?
9. What is the class of control laws which decouple?
10. What are the control laws in decouple in closed-loop tracking?
11. Figure 8 is hard to read and should be described in detail.
12. In conclusions, the authors believe that the proposed method can reduce the influence of nonlinear problem. Do they considered this believe a scientific argument?
Reviewer 2 Report
1- The quality of the English must be improved significantly. Several misused words must be corrected. Those misused words would be misleading and must be corrected before publication. For example, "Pitch angle" must be changed to "elevation angle", "miss distance" must be changed to "tracking error", "the influence of ..." must be changed to "the effect of .." and so on.
2- These authors have published another paper entitled "Rotation matrix error-decoupling methods for Risley prism closed-loop tracking" in the Journal of "Precision Engineering". In that paper, the method of Rotation Matrix Error-Decoupling (RMED) is already well explained. Therefore, instead of repeating the RMED method in this paper, the author could briefly review the RMED method and then focus only on the variable gain function.
3- The author should devise an abbreviation instead of repeating "image-based closed-loop tracking" multiple times. The abbreviations used in their previous paper (such as RMED) and those used in this paper should be consistent.
4- In line 89, it is mentioned that the prisms are the same, so why do the parameters of the formulas after that have the subscripts of 1 and 2? Besides, figure 1 shows the author considered the configuration of tilted-vertical-vertical-tilted double rotating prisms. So, if those prisms were the same, no blind spot would exist. The fact is, there is always a blind spot in a double prism system because those two prisms can not be ultimately the same in practice. The author should make that point in their text.
5- Why is a subscript of 1 in Eqs(4-5)?
6- Instead of repeating the materials already published in your previous paper, please provide more information about the functions used in Fig.5 and Fig.8.
7- Fig.8 is not readable at all. The authors should provide more information about each block (function) used in that figure.
8- Why do we have two G1s in Fig5?
9-What is the unnamed block at the bottom of fig.5?
10- That would be more useful if the authors shared the Simulink files of the paper as supplementary materials.
11- Using theta as the orientation of the prism and azimuth is a bit confusing. It is recommended to use another greek letter (phi, for example) for the prism's orientation.
12- The font of the words of targets 1 to 3 in Eq(29) should be modified for better readability.
13- The results shown in Figures or Tables should not be repeated in the text (like Fig.10, Table 1, and the text below Table 1).
Thank you.
Reviewer 3 Report
Title: Multivariable Decoupling and Nonlinear Correction Method for Image-Based Closed-Loop Tracking of the Risley Prisms System
(1) In Introduction, the authors should mention their previous work on the same topic, for example, “High-precision closed-loop tracking of moving targets based on rotational double prisms, Opt. Eng. 2021, 60(11): 114107”. That work has also proposed a closed-loop tracking method by introducing a controller to solve the coupling relationship and nonlinearity problem for Risley prisms. The authors should address their novelty and contribution in this paper clearly. Any related work from the same group or others should be cited as well.
(2) In Fig. 2, the filed of view of Risley prism system is divided into four regions. How do you define the region with small pitch angle or large pitch angle? Please point this out clearly.
(3) It seems that Section 2 can be optimized to enhance the readability of this paper. As the nonlinearity analysis are referred to [19] and [20], the reader cannot find any new contribution to the analysis method or results.
(4) It is unclear how Eqs. (14) and (15) are obtained. There is no derivation or explanation that can help the reader.
(5) The first-order paraxial approximation method was used in the forward and inverse solutions for Risley prisms. How about the accuracy? What is the influence on the servo control system?
(6) How are the control parameters in Eq. (28) obtained? It is hard for the reader to follow and repeat your work in this way.
(7) The authors performed the moving target tracking experiment in three different regions. What about the performance of the proposed method to track a target moving from one region to another region?
(8) In addition to the steady-state error and dynamic response time, the control system should be further evaluated in terms of the robustness to disturbance and noise.
Round 2
Reviewer 1 Report
The paper was enough improved. I suggest publishing the last version as it is.
Author Response
Response: Thanks very much for taking your time to review this manuscript. I really appreciate all your comments and suggestions! Your comments and suggestions have greatly helped us to improve this manuscript. And thank you very much for your affirmation of our revised manuscript.
Reviewer 2 Report
The authors did what they were supposed to do. I still think Figs 3-4 present nothing new. That relationship is already in Eq(4) and Eq(6). Therefore, Figs 3-4 could be removed.
RMED should be defined just one time in the footnote as "Rotation Matrix Error-Decoupling."
English has significantly improved, but I still think lines 13-14 and 20-23 must be revised for better readability.
That is a good paper, and I would recommend it for publication.
Author Response
Point 1: The authors did what they were supposed to do. I still think Figs 3-4 present nothing new. That relationship is already in Eq(4) and Eq(6). Therefore, Figs 3-4 could be removed.
Response 1: Thank you very much for your suggestion on our manuscript again. According to your suggestion, we have deleted Fig3-4.
Point 2: RMED should be defined just one time in the footnote as "Rotation Matrix Error-Decoupling."
Response 2: Thank you very much for your suggestion. According to your suggestion, we defined the Rotation Matrix Error-Decoupling as RMED when it first appeared in the abstract.
Point 3: English has significantly improved, but I still think lines 13-14 and 20-23 must be revised for better readability.
Response 3: Thank you very much for your advice. We have modified lines 13-14 and 20-23 to increase readability. As follows:
Lines13-14 before modification:
Different from the traditional target tracking equipments, the IBCLT process of the Risley prism is a complex process of double input and double output, and the Risley prism is a nonlinear system.
Lines 13-14 after modification:
Compared with the traditional target tracking equipments, the Risley prism system has two difficulties in the process of IBCLT. First, the Risley prism is a complex coupling system of double input and double output. Second, the Risley prism itself is a nonlinear system.
Lines20-23 before modification:
The IBCLT error deviation of different static targets in the field of view is within 0.025 arcsec, which is 70% lower compared with the fixed gain method; while the steady-state error deviation of moving targets is controlled within 2.5 arcsec, which proves the feasibility and effectiveness of the proposed method.
Lines 20-23 after modification:
The experimental results show that the IBCLT error deviation of different static targets in the field of view is within 0.025 arcsec, which is 70% lower compared with the fixed gain method. Furthermore, the steady-state error deviation of moving targets is controlled within 2.5 arcsec. These experimental results prove the feasibility and effectiveness of the proposed method.
Point 4: That is a good paper, and I would recommend it for publication.
Response 4: Thank you very much for your affirmation of our manuscript. I really appreciate all your comments and suggestions! Your comments and suggestions have greatly helped us to improve this manuscript.
Reviewer 3 Report
The author has revised the original manuscript, and I think the paper can be published after further revising.
Author Response
回应:非常感谢您对我们的稿件提出的意见和建议,这对我们非常有帮助。我们进一步寻求英语专业人士的帮助,以修改手稿中的语言问题。